# Managing minor ailments and pharmacy services: How do people make their decisions?

Léa Charnaux[1,2,3,4], Jérôme Berger[1,2,3,4‡], Clémence Perraudin [1,2,3,4‡]*

1 School of Pharmaceutical Sciences, University of Geneva, Geneva, Switzerland, 2 Institute of Pharmaceutical Sciences of Western Switzerland, University of Geneva, University of Lausanne, Lausanne, Switzerland, 3 Center for Research and Innovation in Clinical Pharmaceutical Sciences, Lausanne University Hospital and University of Lausanne, Lausanne, Switzerland, 4 Department of Ambulatory Care, University Center for Primary Care and Public Health, University of Lausanne, Lausanne, Switzerland

‡ JB and CP are co-last authors of the paper.
* clemence.perraudin@unisante.ch

## Abstract

### Background

Healthcare systems face challenges such as rising costs and workforce shortages. Optimal resource allocation is needed, notably in the management of minor ailments. Community pharmacy services (e.g., minor ailment schemes) remain underused by the population. This study explores the decision-making process when people are managing minor ailments: What influences their choices among the management options? What is their level of information and willingness to use and pay for pharmacy services?

### Methods

A cross-sectional online survey was conducted using a self-administered questionnaire (convenience sampling) from 07.11.2023 to 08.12.2023 in Switzerland. The survey explored general decision-making process in minor ailment management through three clinical scenarios, factors influencing the choice of pharmacy services, as well as public knowledge of three available pharmacy services and their willingness to use and pay for them.

### Results

A total of 508 valid responses were analysed (99.8%). Perceived severity of symptoms (98%, n = 495) and time to deal with symptoms (78%, n = 396) were the most important criteria in making the management decision. The more serious the symptoms were perceived, the less the pharmacy and self-medication were privileged. Choosing pharmacy depended mainly on the perceived staff's skills (83%, n = 420) and direct access to medicines (81%, n = 409). Public awareness of pharmacy

**Data availability statement:** Data was deposited at Unisanté data repository with the survey ID number: 10.16909-dataset-56. Access at: https://doi.org/10.16909/dataset/56.

**Funding:** The survey was carried out as part of a master's thesis in pharmacy at the University of Geneva. Unisanté receives global financial support for the supervision of all master's theses in pharmacy at the University of Geneva from a regional organization, the Société Vaudoise de Pharmacie (SVPh; https://www.svph.ch/). The funder plays no role in the research resulting from these theses. There was no additional external funding received for this study.

**Competing interests:** The authors have declared that no competing interests exist.

services was low. Respondents were more willing to use autonomous prescribing than other services, with low willingness to pay out of their pocket.

## Conclusions

There is a critical gap between the potential demand for minor ailment services in pharmacy and the public's awareness and valuation of them. Actual uptakes could be improved by better communication and uniformization of pharmacy services, as well as the identification of right incentives to achieve the political goal of adequate orientation in the healthcare.

---

## Introduction

While people's choices when managing minor ailments have a direct impact on healthcare systems (availability of professionals, waiting times, cost), their decision-making process is little studied. Minor ailments are defined as "non-complicated medical conditions which can be self-diagnosed and managed, with or without the support of a healthcare professional" [1]. When people must manage minor ailments, multiple options are possible. They can decide to do nothing about the symptoms [2] or to use self-care by dealing with their symptoms by themselves or by asking for advice from some relatives, including or not the use of over-the-counter (OTC) medicines, but without any health professional involvement [2]. Another option is to access the healthcare system through different pathways, e.g., community pharmacy, general practitioner (GP), emergency department (ED), or other professionals according to the healthcare system [2].

Community pharmacies, with their expertise and accessibility, often represent an under-utilised resource in the management of minor ailments [3]. Minor ailment services [4] allow pharmacists to provide essential primary care services in a structured manner; offering information, advice, treatment and/or referral if necessary. They can deliver OTC medicines, as well as autonomous prescribing of "prescription-only medicines" according to the country's agreements and by engaging their responsibilities [5]. Autonomous prescribing is defined as the act that occurs when "a prescriber undertakes prescribing within their scope of practice without the approval or supervision of another health professional" [6,7].

Over the past few years, most healthcare systems have been faced with rising healthcare costs and shortages of healthcare professionals [8,9]. To address these challenges, healthcare systems encourage people with minor ailments to choose appropriate pathways, notably to avoid unnecessary ED and GP consultations [3,10]. Extending the role of community pharmacies is one option for better resource allocation, reducing avoidable ED and GP consultations [3,10]. However, in real life, people can find themselves confronted with numerous possible choices and may have difficulty finding their way around, depending on their individual factors and knowledge of and access to the various alternatives. Healthcare systems need to understand better how people make their choices to target communication campaigns and find the right incentives for effective orientation.

The aim of the study was to explore the criteria influencing the decision-making process when people manage minor ailments; notably the level of information of the public as well as their willingness to use and pay for pharmacy services, notably autonomous prescribing that was extended in 2019 [11]. Switzerland provides an interesting context for a case study with possible transposition to other countries/systems since people who manage minor ailments have all alternatives possible (notably pharmacy services) with a limited influence from the insurance coverage or not. Indeed, in Switzerland, people choose their mandatory health insurance contract with an annual deductible level (i.e., the amount that the insured people have to pay out of pocket for any care before the health insurance starts to cover them), going from 300 Swiss Francs (CHF) to 2'500 Swiss Francs [12] (CHF 1.00 = €1.06 or $1.16) [13]; the insurance premium being lower with a higher deductible level. Generally, most of the 26–40 years old choose the 2'500 Swiss Francs deductibles (58% between 26 and 30 years old) and the older people get, the less they subscribe to this amount of deductible with 24% people between 61 and 65 years old and less than 5% for people over 80 years [14]. Indeed, people in good health will choose a high deductible level and will pay for their use of healthcare services to manage minor ailments directly out of their pocket regardless of the provider or whether the service is covered or not (e.g., pharmacy, GP, ED). In addition, people can also choose restricted insurance models, such as the requirement to consult in priority a selected GP or to contact a telemedicine service to lower their insurance premiums. The case study, therefore provides useful new insights for future studies, as the literature is currently limited almost to the UK context [2,15,16].

## Materials and methods

A cross-sectional survey, reported in accordance with the Checklist for Reporting Of Survey Studies (CROSS) [17], was conducted from November 07, 2023 to December 08, 2023 in the French-speaking part of Switzerland. The questionnaire (S1 File) was constructed from a general model of decision-making and possible options when people must manage minor ailments, created on the basis of published articles (Fig 1) [4,12,18].

The questionnaire included 32 questions (plus 10 sub-questions conditional on participants' answers) divided into three parts. Firstly, socio-demographic characteristics were collected (e.g., gender, age, education, health insurance contract modalities) [19–21]. Secondly, three clinical scenarios that could be managed at the pharmacy (dry cough, diarrhoea, rectal bleeding) [2,20] were presented to explore the general decision-making process in minor ailment management. Respondents were asked to rank preferred management options (e.g., do nothing, consult a GP, self-medication, go to the pharmacy) and assess the level of importance of criteria influencing their decision (e.g., time to deal with symptoms; health insurance model, cost, perceived symptom severity). Thirdly, we explored factors influencing whether or not to choose pharmacy services for the management of minor ailments. Respondents were asked to select which of 12 criteria would prompt them to go to the pharmacy (e.g., geographical proximity, waiting time, consultation room, access to medicines, available services) [22–25]. We also assessed their level of knowledge, willingness to use and pay for three pharmacy services available in Switzerland: 1) autonomous prescribing, 2) prescribing via structured prescribing arrangement and 3) prescription under GP supervision were asked after the three services were defined (S1 Table) [26–29]. For each category of prescription, one service was chosen as an example, but they are not exhaustive of the existing pharmaceutical services.

The research team evaluated and assessed the questionnaire according to the following criteria: 1) face validity (the subjective assessment of the overall impression conveyed by the questionnaire, as well as the clarity of the items and the response methods) and 2) content validity (the subjective evaluation of whether the questions align with the study objectives). The questionnaire was also piloted with four participants from the general population to assess the time required for completion and the level of comprehension. Minor changes were made based on feedback from each stage without assessing test-retest reliability due to the exploratory nature of the questionnaire.

A convenience sample among the general population was recruited using two distribution methods: 1) the questionnaire was posted via social media with an invitation to transfer it to other contacts (snowball sampling method) with two

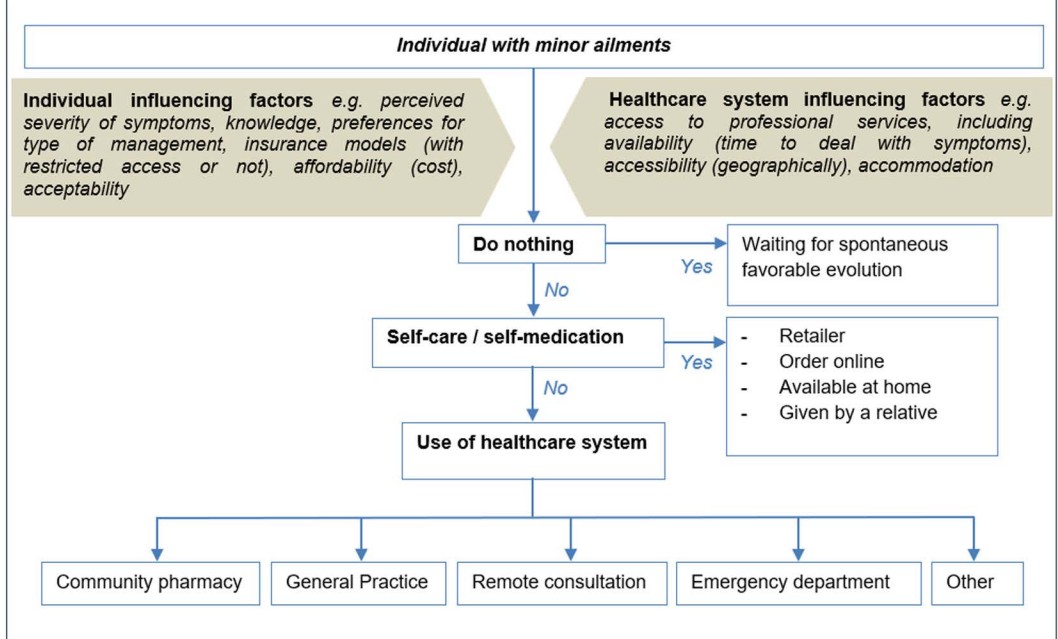

**Fig 1. General model of decision-making process and possible options when people must manage minor ailments.**

recalls; 2) a flyer with a QR code heading towards the questionnaire was distributed in 50 pharmacies, four GP waiting rooms, two ED waiting rooms and a vaccination centre. Inclusion criteria were the age (≥18 years old) and to have a mandatory health insurance contract. No formal exclusion criteria were enacted, and the inclusion and identification of healthcare professionals were considered useful to enable comparison with opinions from the general population.

According to the Swiss Federal Law on Research Involving Human Beings [30], no submission to the Ethics Commission for Research on Human Beings (CER-VD) [31] was required. Tacit consent was obtained during the questionnaire's completion S1 File. The data were collected and managed using the REDCap˚ electronic data capture tool hosted by Vanderbilt University [32]. Descriptive analyses were performed in Microsoft Excel˚. Data was presented as absolute (n)/ relative (%) values for categorical variables and mean/median values for quantitative ones.

## Results

A total of 514 respondents completed the questionnaire. The analyses were conducted on 508 questionnaires after excluding six detected duplicates by Excel duplicate rules. The majority being not-healthcare professionals females with postgraduate degrees in good health and with a restricted insurance model as showed in Table 1.
Preferences for the type of management were different according to the three clinical scenarios that could be managed at the pharmacy (dry cough, diarrhoea, rectal bleeding). The more serious the symptoms were perceived, the less the pharmacy and self-medication were privileged (S2 Table). Depending on the symptom type, respondents chose a pharmacy to get counselling or to buy OTC (S3 Table). The perceived severity of symptoms and the time to deal with symptoms were the two most critical criteria in the decision-making process (Fig 2). Cost was (very) important for only a third of respondents with different deductible amounts.

Respondent's decision about whether or not to go to the pharmacy when dealing with minor ailments depended mainly on the perceived staff's skills (83%, n = 420) and on the direct access to medicines (81%, n = 409). Health insurance coverage was important for only half of the respondents (Fig 3).

**Table 1. Sociodemographic characteristics of respondents (n = 508).**

| Variables | n (%) | Variables | n (%) |
|---|---|---|---|
| **Gender** | | **Age range (years)** | |
| Female | 355 (70%) | 18-50 | 317 (62%) |
| Male | 146 (29%) | 51-64 | 123 (24%) |
| Other | 4 (<1%) | 65-80 | 61 (12%) |
| Unknown | 3 (<1%) | 80+ | 6 (1%) |
| | | Unknown | 1 (<1%) |
| **Education** | | **Health-related profession** | |
| Compulsory schooling | 5 (1%) | Yes | 175 (35%) |
| Secondary vocational | 93 (18%) | Pharmacy | 53 (10%) |
| High school diploma | 40 (8%) | Insurance | 3 (<1%) |
| Professional diploma | 61 (12%) | Other | 115 (23%) |
| University, higher education | 301 (59%) | Unknown | 4 (<1%) |
| Unknown | 6 (1%) | No | 327 (64%) |
| | | Unknown | 6 (1%) |
| **Health insurance model** | | **Annual deductible amount (CHF)** | |
| Restricted models | 287 (56%) | Minimum (300) | 187 (37%) |
| GP first (n = 225) | 225 (44%) | Intermediate (500–2000) | 100 (20%) |
| Telemedicine (n = 26) | 26 (5%) | Maximum (2500) | 184 (36%) |
| HMO* (n = 22) | 22 (4%) | Unknown | 37 (7%) |
| Pharmacy/telephone helpline | 4 (<1%) | | |
| Other | 8 (2%) | | |
| Unknown | 2 (<1%) | | |
| Unrestricted model | 201 (40%) | | |
| Unknown | 20 (4%) | | |
| **Referent GP** | | **Referent pharmacy** | |
| Yes | 453 (89%) | Yes | 290 (57%) |
| No | 50 (10%) | Multiples pharmacies | 92 (18%) |
| Unknown | 5 (1%) | No (always change/ no need) | 122 (24%) |
| | | Unknown | 4 (<1%) |
| **Perceived health status** | | **Chronic illness** | |
| Excellent | 85 (17%) | Yes | 121 (24%) |
| Very good | 171 (33%) | No | 370 (73%) |
| Good | 201 (40%) | Multiple chronic illnesses | 9 (2%) |
| Unsatisfying | 37 (7%) | Unknown | 6 (1%) |
| Poor | 9 (2%) | | |
| Unknown | 5 (1%) | | |

*Health Maintenance Organisation.

The pharmacy services were little known by the public as well as health care professionals (excluding pharmacy-related) (Fig 4). Respondents were more willing to use autonomous prescribing than other services (Fig 5), with low willingness to pay out of their pocket. Only half and third of the respondents were willing to pay the recommended price for the services "autonomous prescribing" (15–20 Swiss francs) and "prescribing via structured prescribing arrangement" (30–40 Swiss francs), respectively. Willingness to pay for "prescribing under GP supervision" was not questioned because the service is covered by health insurance.

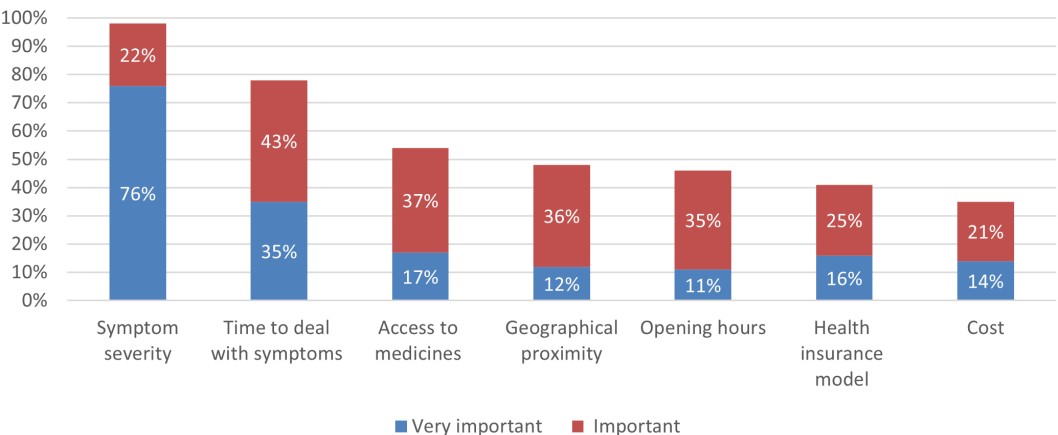

**Fig 2. Criteria considered as (very) important in the decision-making process to choose the type of management preferred (Question When you answered for the three scenarios above, how did you come to your decision? What criteria did you consider?).**

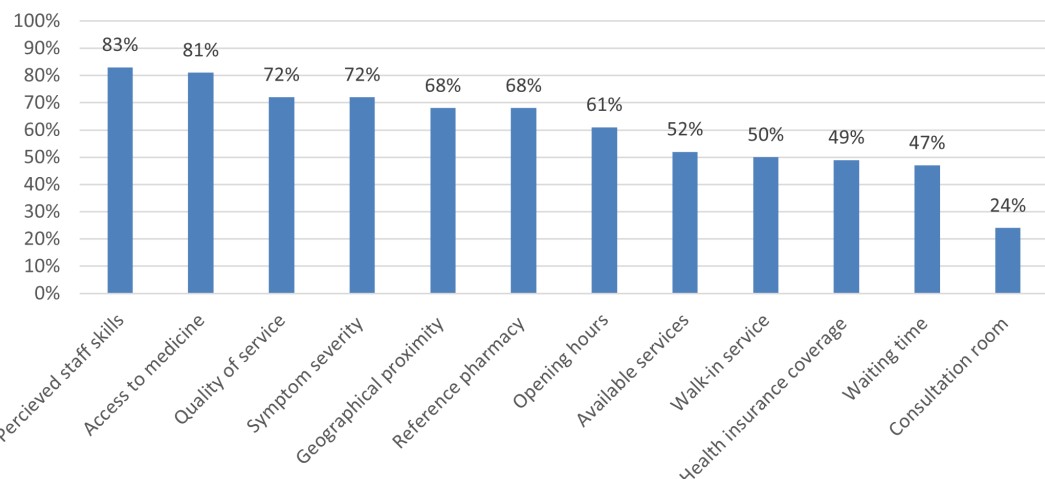

**Fig 3. Criteria influencing respondents in their decision to go or not to the pharmacy (Question: What criteria do you use to decide whether or not to go to the pharmacy when you are dealing with minor ailments?).**

## Discussion

The present study explored the decision-making process when people are managing minor ailments and their willingness to use and pay for pharmacy services available in Switzerland. The study showed that perceived severity of symptoms (98%, n = 495) and time to deal with symptoms (78%, n = 396) were the most important criteria in making the management decision. They are willing to go to pharmacy, mainly driven by the perceived staff's skills (183%, n = 420) and direct access to medicines (81%, n = 409). Public awareness of pharmacy services was low. Respondents were more willing to use autonomous prescribing than other services, with low willingness to pay out of their pocket.

## Comparison of the literature

To our knowledge, this study is the first to explore public opinion on the management of minor ailments in Switzerland, and only a few studies are available limited almost to the UK context. The subjective perceived severity of symptoms was

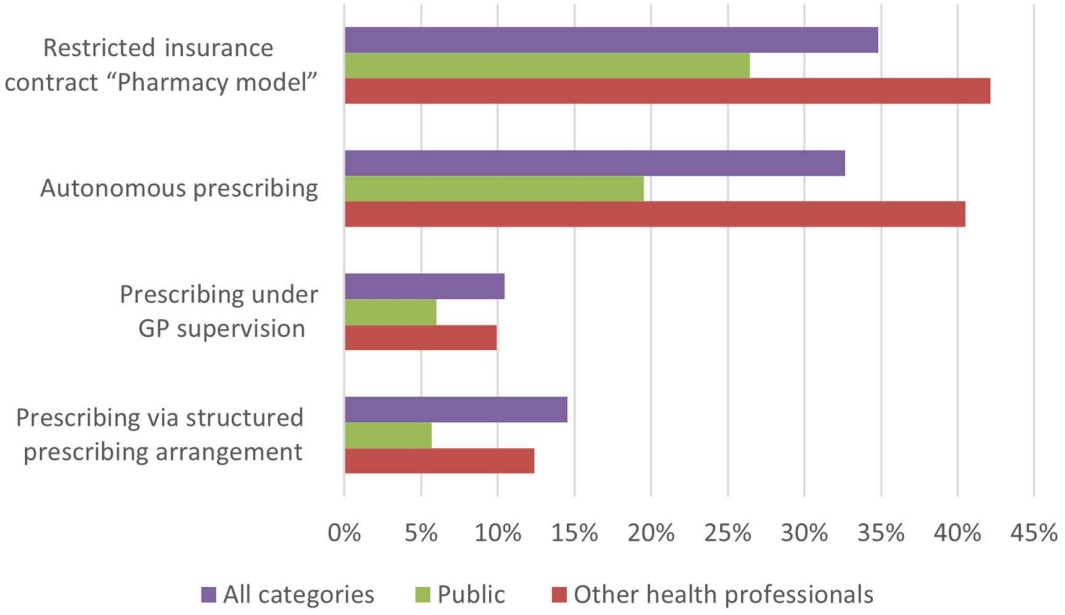

**Fig 4. Levels of knowledge of pharmacy services.**

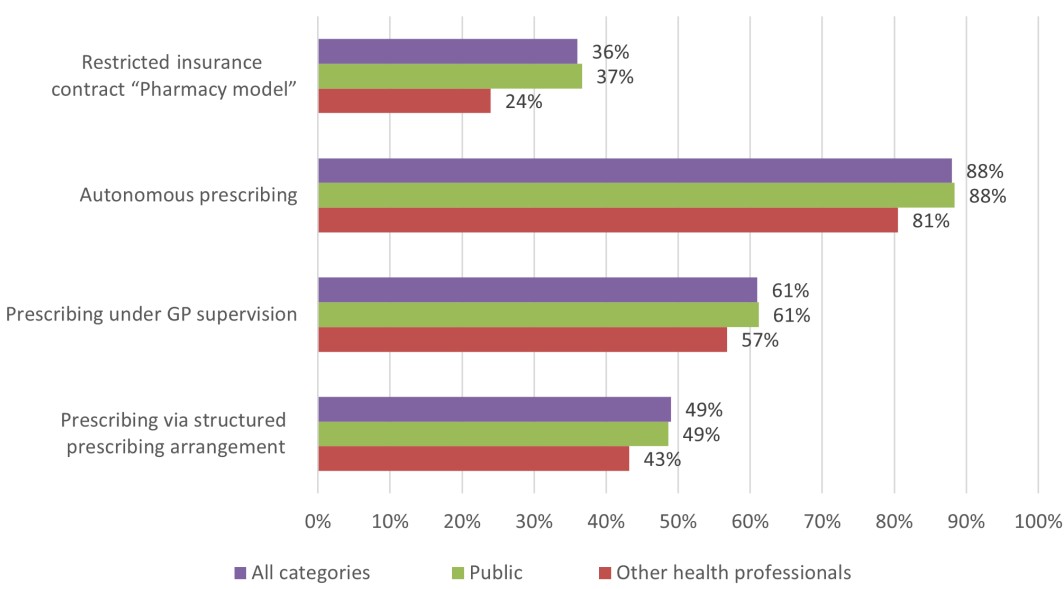

**Fig 5. Willingness to use pharmacy services.**

the most important criterion in the respondents' decision-making process, as a previous study showed [15]. Moreover, "perceived severity of symptoms" is an important criterion for going to the pharmacy (Fig 3) and the more severe the symptoms were perceived, the less self-medication and pharmacy preferred (S2 Table). These results corroborate previous studies stating that pharmacy and self-medication are mostly chosen when the symptoms are considered too light to consult a GP [33]. This could be due to a knowledge gap on pharmacy services and on pharmacists' competencies as the

three clinical scenarios could be managed in pharmacy (i.e., absence of red flags suggesting the need for GP consultation) [34] and more than 1,500 pharmacists have complementary training certificate in primary care anamnesis [35]. This knowledge gap has also been highlighted as a challenge in England [3].

Time to deal with symptoms was also (very) important for most of the respondents and could be linked with the perceived severity of symptoms, supposing that the more severe the symptoms were perceived, the faster resolution of symptoms was sought. The literature also shows a trade-off between time to deal with symptoms and willingness to pay additional costs [15]. Contrary to other studies [2,15,16,24], the cost and the health insurance coverage did not seem to be important in the decision, probably linked to the deductible system in Switzerland since, ultimately, most people will pay for any services out of their pocket in case of minor ailments.

The perceived community pharmacy staff's skills and quality of the services were important criteria for respondents in deciding whether or not to go to the pharmacy. This corroborates previous results where respondents stated the need to know that pharmacists were qualified to manage minor ailments [16,36], the need to trust the pharmacy's staff skills [33,37] and to get a better understanding of the symptoms and their management after visiting a pharmacy [16]. Additionally, the people who visited pharmacies frequently seemed to be more willing to use minor ailment services. This could be related to good relationships with pharmacists and enhanced public awareness to their competencies [24]. Moreover, direct access to medicines was an important criterion in deciding whether or not to go to the pharmacy, and self-medication was always chosen before the pharmacy visit in each clinical scenario. This suggests that the public perceives pharmacies mainly as places where OTC medicines are bought and less as for triaging [2,38]. Indeed, the public and health professionals (excluding pharmacy-related) were only poorly aware of pharmacy services. Better communication and visibility are needed as knowledge and awareness are crucial, i.e., because recipients cannot choose pharmacy services if they do not know them. A systematic review has highlighted these facts involving the lack of use of pharmacy services to their fullest potential [39] therefore, a communication campaign was initiated by the professional body for pharmacists Pharma Suisse [40]. In Switzerland, due to federalism and economic liberalism in the healthcare system, there is a diversity of pharmacy services to manage minor ailments with different characteristics (S1 Table), driven by different stakeholders (e.g., health insurers, PharmaSuisse, health authorities). This diversity also makes it difficult for the public to understand what services are available when to use them and make the best possible choice according to the personal importance attached to each criterion. Uniformed pharmacy services to manage minor ailments and communication supported by stakeholders would certainly enable sustaining services and grow their use and impact. For example, the NHS follows a national service strategy while maintaining some regional specificities [41] because isolated initiatives tend to be more ephemerous and struggle with sustainable funding [3]. Scotland has already used a national service for 20 years, enabling equal national service access for all eligible persons [3,42].

Willingness to use pharmacy services are relatively high but with low willingness to pay out of their pocket. In Switzerland, pharmacy services are rarely covered [3,23], and medicines costs are only covered with a prescription [43]. Moreover, the pricing of pharmacy services differs from one pharmacy to another whilst their coverage depends on the health insurance model, which could result in poor public knowledge. The willingness to pay for pharmacy services is a barrier to the use of pharmacy services [4,44], and the Swiss population has not been used to pay for services in pharmacy, except for vaccination services [45]. The coverage of pharmacy services could be a good incentive to improve their use as it is done in the UK or Canada in minor ailments services [3,44]. Some countries, like Scotland, New Zealand and France, even go further by covering the costs of the medicines provided under minor ailments services [3]. In Switzerland, restricted insurance contracts "pharmacy model" covering pharmacy services for minor ailments services are developing in the face of rising insurance premiums. But these contracts are new and less known by the public. They compete with more popular models such as the "contact a selected GP first" and HMO models. Although a third of respondents would be willing to subscribe to such restricted insurance contracts "pharmacy model", the impact in terms of better use of the

healthcare system may not be guaranteed, as implementation of the services in community pharmacies remains low (S1 Table) and "pharmacy model" contracts are always mixed: people could choose other available options for minor ailments services, such as telephone helpline. Furthermore, since insureds can change insurance every year in Switzerland, it is difficult to anticipate the impact of these models on public behavior.

## Strengths and limits

This protocol is the first of its kind in Switzerland and provides an initial insight into people's opinions and the decision-making process regarding the management of minor ailments in this population. The interest of this study is to examine these decision-making processes in a different context from other countries than the UK, Australia or Canada, where most studies on community pharmacy and minor ailment services are conducted. In Switzerland, due to the deductible system, most healthy adults using a health service to treat a minor ailment would pay for the service out of their own pocket, regardless of the provider. We believe that limiting the influence of the cost factor provides interesting results that can be transposed to other contexts.

As for the study limitations, an accessibility bias exists as the questionnaire was only available online and in French. No preventative measures were taken to avoid multiple completions by the same individual. However, an analysis of duplicates was conducted following the collection of all questionnaires. The sample was not representative of the Swiss population as the study was exploratory. Indeed, the basic insurance contracts were overrepresented (40% vs 22%) [45]. People having restricted insurance model « Pharmacy model » (7% vs 29% for the 19–26 years old and 28% for the > 26 years old) [45], as well as extreme deductibles (300 Swiss francs: 37% vs 46%; 2'500 Swiss francs: 36% vs 41%) [46] were under-represented. There was also a possible over-representation of young people [47]. A bias of desirability in favor of pharmacy services was also possible as the questionnaire was distributed in pharmacies but was minimized by having an online self-administered questionnaire. Furthermore, the QR code mostly reached persons that were already coming in pharmacies and possibly more aware of pharmacy services existence. Social media distribution helps to compensate for this bias. It is also to consider that the answers related to pharmacy services depended on how we defined them in the questionnaire. Robust quantitative findings by discrete choice experiments in Switzerland would enable a better understanding of the decision-making process and preferences for the public (e.g., possible trade-off and willingness to pay) in a context that has limited external factors impacting the decisions due to the free choice of care and the out of the pocket costs so that incentives to achieve public health objectives can be aligned with them.

Understanding how individuals make decisions and express preferences in managing minor ailments is essential for achieving public health goals. For researchers, this paper provides a foundation for designing studies that better reflect real-world behaviors and inform evidence-based interventions. In Switzerland, pharmacists and policymakers should do more to communicate the fact that a health service to treat a minor ailment is in most cases paid for out-of-pocket by patients, and that treatment in a pharmacy costs less than other alternatives while limiting waiting times and providing direct access to a drug if necessary. Finally, pharmacies should effectively implement the services and successfully capture potential demand.

## Conclusions

The paper shows a critical gap between the potential demand for minor ailment services in pharmacy and the public's awareness and valuation of them in Switzerland, but which is potentially just as true in many countries. Actual uptakes could be improved by better communication and design/uniformization of pharmacy services, as well as find the right incentives and strategies to guide the choice of the pharmacy. Future research needs to be conducted nationally to find the effective combination of multi-level and multi-dimensional implementation strategies that will achieve public health goals.

## Supporting information

**S1 File. Questionnaire "Exploration of the opinions in the management of minor ailments".**
(S1 File.PDF)

**S1 Table. Examples of pharmacy services to manage minor ailments available in Switzerland.**
(S1 Table.DOCX)

**S2 Table. Orientation decision based on perceived severity of symptoms.**
(S2 Table.DOCX)

**S3 Table. Results for the question: "***Indicate the minor ailments that would make you visit a pharmacy (yes I go to the pharmacy for a consultation or a counselling, yes I go to the pharmacy to buy some medicines without prescriptions, no I don't go to the pharmacy if I have these symptoms, unapplicable, I don't know, I don't want to answer"***.**
(S3 Table.DOCX)

## Acknowledgments

We thank all participants for their time and their commitment.

## Author contributions

**Conceptualization:** Léa Charnaux, Jérôme Berger, Clémence Perraudin.

**Data curation:** Léa Charnaux, Clémence Perraudin.

**Formal analysis:** Léa Charnaux.

**Funding acquisition:** Jérôme Berger.

**Investigation:** Léa Charnaux.

**Methodology:** Léa Charnaux, Clémence Perraudin.

**Project administration:** Léa Charnaux, Jérôme Berger, Clémence Perraudin.

**Resources:** Léa Charnaux.

**Software:** Léa Charnaux.

**Supervision:** Jérôme Berger, Clémence Perraudin.

**Validation:** Jérôme Berger, Clémence Perraudin.

**Visualization:** Léa Charnaux, Clémence Perraudin.

**Writing – original draft:** Léa Charnaux.

**Writing – review & editing:** Jérôme Berger, Clémence Perraudin.

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
