## [Decision Letter · Decision Letter 0]

5 May 2025

PONE-D-25-05729Managing minor ailments and pharmacy services: how do people make their decisions?PLOS ONE

Dear Dr. Perraudin,

Thank you for submitting your manuscript to PLOS ONE. After careful consideration, we feel that it has merit but does not fully meet PLOS ONE’s publication criteria as it currently stands. Therefore, we invite you to submit a revised version of the manuscript that addresses the points raised during the review process.

**We have now completed the evaluation of your manuscript. Based on the feedback received from the peer reviewers** , we have concluded that your manuscript requires **minor revisions** before it can be further considered for publication in PLOS ONE.

**Please revise the manuscript to address all comments raised by the reviewers.** Kindly note that **Reviewer 2 has provided their comments in an uploaded PDF file** . Additionally, please revise the percentages reported in **Table 1** . Some values listed as 1% are actually **less than 1%** (e.g., gender), and the current presentation results in a total exceeding 100%.

We look forward to receiving your revised manuscript.

Kind regards,

Mohammad Nusair, Ph.D

Academic Editor

PLOS ONE

**Journal Requirements:**

1. When submitting your revision, we need you to address these additional requirements. Please ensure that your manuscript meets PLOS ONE's style requirements, including those for file naming. The PLOS ONE style templates can be found at https://journals.plos.org/plosone/s/file?id=wjVg/PLOSOne_formatting_sample_main_body.pdf and https://journals.plos.org/plosone/s/file?id=ba62/PLOSOne_formatting_sample_title_authors_affiliations.pdf 2. Thank you for stating in your Funding Statement: All the master’s thesis in pharmacy at the University of Geneva and conducted at Unisanté are financially supported by a regional organisation, the Community Pharmacist Association in Vaud region (SVPh or Société Vaudoise de Pharmacie). Please provide an amended statement that declares *all* the funding or sources of support (whether external or internal to your organization) received during this study, as detailed online in our guide for authors at http://journals.plos.org/plosone/s/submit-now.  Please also include the statement “There was no additional external funding received for this study.” in your updated Funding Statement. Please include your amended Funding Statement within your cover letter. We will change the online submission form on your behalf. 3. We note that you have indicated that there are restrictions to data sharing for this study. For studies involving human research participant data or other sensitive data, we encourage authors to share de-identified or anonymized data. However, when data cannot be publicly shared for ethical reasons, we allow authors to make their data sets available upon request. For information on unacceptable data access restrictions, please see http://journals.plos.org/plosone/s/data-availability#loc-unacceptable-data-access-restrictions.  Before we proceed with your manuscript, please address the following prompts: a) If there are ethical or legal restrictions on sharing a de-identified data set, please explain them in detail (e.g., data contain potentially identifying or sensitive patient information, data are owned by a third-party organization, etc.) and who has imposed them (e.g., a Research Ethics Committee or Institutional Review Board, etc.). Please also provide contact information for a data access committee, ethics committee, or other institutional body to which data requests may be sent. b) If there are no restrictions, please upload the minimal anonymized data set necessary to replicate your study findings to a stable, public repository and provide us with the relevant URLs, DOIs, or accession numbers. Please see http://www.bmj.com/content/340/bmj.c181.long for guidelines on how to de-identify and prepare clinical data for publication. For a list of recommended repositories, please see https://journals.plos.org/plosone/s/recommended-repositories. You also have the option of uploading the data as Supporting Information files, but we would recommend depositing data directly to a data repository if possible. Please update your Data Availability statement in the submission form accordingly. 4. Your ethics statement should only appear in the Methods section of your manuscript. If your ethics statement is written in any section besides the Methods, please move it to the Methods section and delete it from any other section. Please ensure that your ethics statement is included in your manuscript, as the ethics statement entered into the online submission form will not be published alongside your manuscript. 5. We note that this data set consists of interview transcripts. Can you please confirm that all participants gave consent for interview transcript to be published? If they DID provide consent for these transcripts to be published, please also confirm that the transcripts do not contain any potentially identifying information (or let us know if the participants consented to having their personal details published and made publicly available). We consider the following details to be identifying information:- Names, nicknames, and initials- Age more specific than round numbers- GPS coordinates, physical addresses, IP addresses, email addresses- Information in small sample sizes (e.g. 40 students from X class in X year at X university)- Specific dates (e.g. visit dates, interview dates)- ID numbers Or, if the participants DID NOT provide consent for these transcripts to be published:- Provide a de-identified version of the data or excerpts of interview responses- Provide information regarding how these transcripts can be accessed by researchers who meet the criteria for access to confidential data, including:a) the grounds for restrictionb) the name of the ethics committee, Institutional Review Board, or third-party organization that is imposing sharing restrictions on the datac) a non-author, institutional point of contact that is able to field data access queries, in the interest of maintaining long-term data accessibility.d) Any relevant data set names, URLs, DOIs, etc. that an independent researcher would need in order to request your minimal data set. For further information on sharing data that contains sensitive participant information, please see: https://journals.plos.org/plosone/s/data-availability#loc-human-research-participant-data-and-other-sensitive-data If there are ethical, legal, or third-party restrictions upon your dataset, you must provide all of the following details (https://journals.plos.org/plosone/s/data-availability#loc-acceptable-data-access-restrictions):a) A complete description of the datasetb) The nature of the restrictions upon the data (ethical, legal, or owned by a third party) and the reasoning behind themc) The full name of the body imposing the restrictions upon your dataset (ethics committee, institution, data access committee, etc)d) If the data are owned by a third party, confirmation of whether the authors received any special privileges in accessing the data that other researchers would not havee) Direct, non-author contact information (preferably email) for the body imposing the restrictions upon the data, to which data access requests can be sent

Reviewers' comments:

Reviewer's Responses to Questions

**Comments to the Author**

1. Is the manuscript technically sound, and do the data support the conclusions?

Reviewer #1: Yes

Reviewer #2: Yes

2. Has the statistical analysis been performed appropriately and rigorously? 

Reviewer #1: Yes

Reviewer #2: Yes

3. Have the authors made all data underlying the findings in their manuscript fully available?

Reviewer #1: Yes

Reviewer #2: Yes

4. Is the manuscript presented in an intelligible fashion and written in standard English?

Reviewer #1: Yes

Reviewer #2: Yes

5. Review Comments to the Author

**Reviewer #1: ** It's interesting to learn about this through a case study conducted in a country other than the UK, Australia, or Canada, where most studies on community pharmacy and minor ailment services are conducted. I also think the findings of this paper are applicable to other countries, since although people are willing to use pharmacy services to manage minor ailments, they are unaware of their existence and have a low willingness to pay.

In the Methods section (line 122), the authors state that the questionnaire used was evaluated and “validated” by the research team. I don't think the word "validated" is correct, since in my opinion, validating a questionnaire requires assessing both reliability (something the authors do when evaluating Face and Content validity) and feasibility (alpha Cronbach or any other test). Therefore, I would modify the text ("The research team evaluated and “assessed” the questionnaire according to the...").

In Results, I think the text should be clarified a bit. The authors say (lines 144-145): “They were not healthcare professionals and had a restricted insurance model with a minimum annual deductible, as shown in Table 1”. In reality, those who had "restricted models" were 56%, which, although it is the majority, is a percentage close to 50%. In my opinion, it does not justify the generalization used since it begins with "They were." I think it would be better to start with “the majority being non-healthcare professionals”.

It is interesting to see that, although as described in table S2, the more serious the symptoms were, the less self-medication and pharmacy were preferred, in Figure 3, “perceived severity” is the fourth cause for going to the pharmacy. This should surely be discussed in the Discussion because at first glance it does not make sense. In fact, the discussion says (line 183) that “The more severe the symptoms were perceived, the less self-medication and pharmacy preferred”, but it does not add that the severity of symptoms was the fourth cause for going to the pharmacy.

(Line 219) In line 219 when the authors state that “there is a diversity of pharmacy services with different characteristics” I would clarify that they refer to “different services to manage minor ailments” to avoid confusion with other different services such as Medication review, adherence, reconciliation and so on.

Regarding REFERENCES, authors should always follow the same criteria when describing the journal: either use the full name, as in Ref#2, or use the abbreviated name, as they do in Ref#14:

Ref#1. A semicolon should follow London, before the year.

Ref#2. The volume is missing: BJGP 2006; 56(533):911-7

Ref#5. Delete “Nov”

Ref#6. Delete “Dec”

Ref#8. They should include a link to the website.

Ref#10. Delete “Fevr”

Ref#13. Delete “Aut”

Ref#14. Delete “24 Janv”

Ref#17. A link to the website should be included.

Ref#19. Delete “Oct”

Ref#31. Delete “Jull”

Ref#34. I would reference this as: Rutter P. Community pharmacy symptoms, diagnosis and treatment. 5th edition. London: Elsevier ed.; 2020. 416 p.

Ref#34. A link to the website should be included.

Ref#43. Or, Int J Gen Med, or include the full name of the journal. Volume (2016;9:291-302).

Ref#44. Basel is missing from the journal name, and delete "Nov 29": Pharmacy (Basel). 2018:6(4):125.

Ref#45. A link to the website should be included.

**Reviewer #2:**  Dear Authors,

Please refer to my comments in the attached file.

Kind regards,

Dear Authors,

Please refer to my comments in the attached file.

Kind regards,

Dear Authors,

Please refer to my comments in the attached file.

Kind regards,

6. PLOS authors have the option to publish the peer review history of their article (what does this mean? ). If published, this will include your full peer review and any attached files.

**Do you want your identity to be public for this peer review?** For information about this choice, including consent withdrawal, please see our Privacy Policy .

Reviewer #1: **Yes: ** miguel angel gastelurrutia

Reviewer #2: No

---

## [Author Response · Author response to Decision Letter 1]

19 Jun 2025

Journal Requirements:

Additionally, please revise the percentages reported in Table 1. Some values listed as 1% are actually less than 1% (e.g., gender), and the current presentation results in a total exceeding 100%. Thanks for the comment. Percentages in the table have been rounded off and values below 1 noted as “<1%” to correspond as closely as possible to 100%.

All the master’s thesis in pharmacy at the University of Geneva and conducted at Unisanté are financially supported by a regional organisation, the Community Pharmacist Association in Vaud region (SVPh or Société Vaudoise de Pharmacie).

The funding statement has been changed (see below). The organisation does not financially support author individually and no grant number is associated to the research as the support is general. Funding statement : “The survey was carried out as part of a master's thesis in pharmacy at the University of Geneva. Unisanté receives global financial support for the supervision of all master's theses in pharmacy at the University of Geneva from a regional organization, the Société Vaudoise de Pharmacie (SVPh; https://www.svph.ch/). The funder plays no role in the research resulting from these theses. There was no additional external funding received for this study.”

Thanks to this comment. After checking with our colleague at the institutional data repository, the dataset is now available in open access. Five columns with text comments have been removed for a de-identifying dataset.

Access request can be made by research teams at : https://doi.org/10.16909/dataset/56

Survey ID number : 10.16909-dataset-56

4. Your ethics statement should only appear in the Methods section of your manuscript. If your ethics statement is written in any section besides the Methods, please move it to the Methods section and delete it from any other section. Please ensure that your ethics statement is included in your manuscript, as the ethics statement entered into the online submission form will not be published alongside your manuscript. Ethics statement has been moved to Materials and methods section (page 7 lines 141-146)

5. We note that this data set consists of interview transcripts. Can you please confirm that all participants gave consent for interview transcript to be published? No, data only include response to questionnaire S1 file.

If they DID provide consent for these transcripts to be published, please also confirm that the transcripts do not contain any potentially identifying information (or let us know if the participants consented to having their personal details published and made publicly available). We consider the following details to be identifying information:

- Names, nicknames, and initials

- Age more specific than round numbers

- GPS coordinates, physical addresses, IP addresses, email addresses

- Information in small sample sizes (e.g. 40 students from X class in X year at X university)

- Specific dates (e.g. visit dates, interview dates)

- ID numbers

Or, if the participants DID NOT provide consent for these transcripts to be published:

- Provide a de-identified version of the data or excerpts of interview responses

- Provide information regarding how these transcripts can be accessed by researchers who meet the criteria for access to confidential data, including:

a) the grounds for restriction

b) the name of the ethics committee, Institutional Review Board, or third-party organization that is imposing sharing restrictions on the data

c) a non-author, institutional point of contact that is able to field data access queries, in the interest of maintaining long-term data accessibility.

d) Any relevant data set names, URLs, DOIs, etc. that an independent researcher would need in order to request your minimal data set.

For further information on sharing data that contains sensitive participant information, please see: https://journals.plos.org/plosone/s/data-availability#loc-human-research-participant-data-and-other-sensitive-data

If there are ethical, legal, or third-party restrictions upon your dataset, you must provide all of the following details (https://journals.plos.org/plosone/s/data-availability#loc-acceptable-data-access-restrictions):

a) A complete description of the dataset

b) The nature of the restrictions upon the data (ethical, legal, or owned by a third party) and the reasoning behind them

c) The full name of the body imposing the restrictions upon your dataset (ethics committee, institution, data access committee, etc)

d) If the data are owned by a third party, confirmation of whether the authors received any special privileges in accessing the data that other researchers would not have

e) Direct, non-author contact information (preferably email) for the body imposing the restrictions upon the data, to which data access requests can be sent

6. Please review your reference list to ensure that it is complete and correct. If you have cited papers that have been retracted, please include the rationale for doing so in the manuscript text, or remove these references and replace them with relevant current references. Any changes to the reference list should be mentioned in the rebuttal letter that accompanies your revised manuscript. If you need to cite a retracted article, indicate the article’s retracted status in the References list and also include a citation and full reference for the retraction notice. The reference list has been only modified to Vancouver format according to author instructions.

Comments to the Author

Reviewer's Responses to Questions

1. Is the manuscript technically sound, and do the data support the conclusions?

Reviewer #1: Yes

Reviewer #2: Yes

2. Has the statistical analysis been performed appropriately and rigorously?

Reviewer #1: Yes

Reviewer #2: Yes

3. Have the authors made all data underlying the findings in their manuscript fully available?

Reviewer #1: Yes

Reviewer #2: Yes

4. Is the manuscript presented in an intelligible fashion and written in standard English?

Reviewer #1: Yes

Reviewer #2: Yes

5. Review Comments to the Author

Reviewer #1: It's interesting to learn about this through a case study conducted in a country other than the UK, Australia, or Canada, where most studies on community pharmacy and minor ailment services are conducted. I also think the findings of this paper are applicable to other countries, since although people are willing to use pharmacy services to manage minor ailments, they are unaware of their existence and have a low willingness to pay. Thanks for your comment, we also believe that publishing illustrations outside the UK, Australia and Canada is important for international literature. Linking to a comment from reviewer 2, we have added this to the strengths of the study: “This protocol is the first of its kind in Switzerland and provides an initial insight into people’s opinions and the decision-making process regarding the management of minor ailments in this population. The interest of this study is to examine these decision-making processes in a different context from other countries than the UK, Australia or Canada, where most studies on community pharmacy and minor ailment services are conducted. In Switzerland, due to the deductible system, most healthy adults using a health service to treat a minor ailment would pay for the service out of their own pocket, regardless of the provider. We believe that limiting the influence of the cost factor provides interesting results that can be transposed to other contexts.” (discussion section, page14 lines 266-273)

In the Methods section (line 122), the authors state that the questionnaire used was evaluated and “validated” by the research team. I don't think the word "validated" is correct, since in my opinion, validating a questionnaire requires assessing both reliability (something the authors do when evaluating Face and Content validity) and feasibility (alpha Cronbach or any other test). Therefore, I would modify the text ("The research team evaluated and “assessed” the questionnaire according to the..."). The sentence has been modified as suggested (section materials and methods page 7 lines 125-128).

In Results, I think the text should be clarified a bit. The authors say (lines 144-145): “They were not healthcare professionals and had a restricted insurance model with a minimum annual deductible, as shown in Table 1”. In reality, those who had "restricted models" were 56%, which, although it is the majority, is a percentage close to 50%. In my opinion, it does not justify the generalization used since it begins with "They were." I think it would be better to start with “the majority being non-healthcare professionals”. You are perfectly right. The sentence has been modified as suggested: “The majority being not-healthcare professionals females with postgraduate degrees in good health and with a restricted insurance model as showed in Table 1.” (section results, page 8 lines 148-150).

It is interesting to see that, although as described in table S2, the more serious the symptoms were, the less self-medication and pharmacy were preferred, in Figure 3, “perceived severity” is the fourth cause for going to the pharmacy. This should surely be discussed in the Discussion because at first glance it does not make sense. In fact, the discussion says (line 183) that “The more severe the symptoms were perceived, the less self-medication and pharmacy preferred”, but it does not add that the severity of symptoms was the fourth cause for going to the pharmacy. We do not interpret these two results as ambivalent. Indeed, Figure 3 highlights the fact that “perceived severity of symptoms” is an important criterion in determining whether or not to go to the pharmacy. However, this result does not directly indicate whether people are more likely to visit a pharmacy if their symptoms are more or less severe, since the question is not formulated in this way. What's interesting to note here is that this criterion is the first in the ranking not to be linked to the characteristics of a particular pharmacy. It is the elements presented in S2 that allow us to interpret that the more severe the symptoms, the less likely people are to go to the pharmacy. We added the precision on the discussion section (page 11 lines 199-202): “Moreover, “perceived severity of symptoms” is an important criterion for going to the pharmacy (Fig 3) and the more severe the symptoms were perceived, the less self-medication and pharmacy preferred (S2 Table).”

(Line 219) In line 219 when the authors state that “there is a diversity of pharmacy services with different characteristics” I would clarify that they refer to “different services to manage minor ailments” to avoid confusion with other different services such as Medication review, adherence, reconciliation and so on. Thanks, we have added the adding as suggested to precise (section discussion page 12 lines 232-235)

Regarding REFERENCES, authors should always follow the same criteria when describing the journal: either use the full name, as in Ref#2, or use the abbreviated name, as they do in Ref#14:

Ref#1. A semicolon should follow London, before the year.

Ref#2. The volume is missing: BJGP 2006; 56(533):911-7

Ref#5. Delete “Nov”

Ref#6. Delete “Dec”

Ref#8. They should include a link to the website.

Ref#10. Delete “Fevr”

Ref#13. Delete “Aut”

Ref#14. Delete “24 Janv”

Ref#17. A link to the website should be included.

Ref#19. Delete “Oct”

Ref#31. Delete “Jull”

Ref#34. I would reference this as: Rutter P. Community pharmacy symptoms, diagnosis and treatment. 5th edition. London: Elsevier ed.; 2020. 416 p.

Ref#34. A link to the website should be included.

Ref#43. Or, Int J Gen Med, or include the full name of the journal. Volume (2016;9:291-302).

Re

---

## [Decision Letter · Decision Letter 1]

11 Jul 2025

PONE-D-25-05729R1Managing minor ailments and pharmacy services: how do people make their decisions?PLOS ONE

Dear Dr. Perraudin,

Thank you for submitting your manuscript to PLOS ONE. After careful consideration, we feel that it has merit but does not fully meet PLOS ONE’s publication criteria as it currently stands. Therefore, we invite you to submit a revised version of the manuscript that addresses the points raised during the review process.

All reviewers have confirmed that the revisions made in response to their comments are satisfactory, and they support the acceptance of your manuscript. However, one reviewer noted a minor typographical error that should be corrected before final acceptance. Accordingly, we are issuing a **decision of minor revisions** to allow you to correct this issue.

We look forward to receiving your revised manuscript.

Kind regards,

Mohammad Nusair, Ph.D

Academic Editor

PLOS ONE

Journal Requirements:

Reviewers' comments:

Reviewer's Responses to Questions

**Comments to the Author**

1. If the authors have adequately addressed your comments raised in a previous round of review and you feel that this manuscript is now acceptable for publication, you may indicate that here to bypass the “Comments to the Author” section, enter your conflict of interest statement in the “Confidential to Editor” section, and submit your "Accept" recommendation.

Reviewer #1: All comments have been addressed

Reviewer #2: All comments have been addressed

2. Is the manuscript technically sound, and do the data support the conclusions?

Reviewer #1: Yes

Reviewer #2: Yes

3. Has the statistical analysis been performed appropriately and rigorously? 

Reviewer #1: Yes

Reviewer #2: Yes

4. Have the authors made all data underlying the findings in their manuscript fully available?

Reviewer #1: Yes

Reviewer #2: Yes

5. Is the manuscript presented in an intelligible fashion and written in standard English?

Reviewer #1: Yes

Reviewer #2: Yes

6. Review Comments to the Author

Reviewer #1: My only comment is that in line 118, the verb "assess" should be in past tense (Assesssed)

[118] access to medicines, available services) [22-25]. We also ASSESSED their level of knowledge,

Reviewer #2: Dear Author,

Thank you for addressing all my comments. I have no further comment.

Regards,

Dear Author,

Thank you for addressing all my comments. I have no further comment.

Regards,

7. PLOS authors have the option to publish the peer review history of their article (what does this mean? ). If published, this will include your full peer review and any attached files.

**Do you want your identity to be public for this peer review?** For information about this choice, including consent withdrawal, please see our Privacy Policy .

Reviewer #1: No

Reviewer #2: No

---

## [Author Response · Author response to Decision Letter 2]

4 Aug 2025

We have corrected the typographical error asked by the reviewer and converted the figures in the good format.

Everything seems to be in order for publication, and we thank you for that.

Sincerely,

Dr. Clémence Perraudin & colleagues

---

## [Editor Report · Decision Letter 2]

6 Aug 2025

Managing minor ailments and pharmacy services: how do people make their decisions?

PONE-D-25-05729R2

Dear Dr. Perraudin,

We’re pleased to inform you that your manuscript has been judged scientifically suitable for publication and will be formally accepted for publication once it meets all outstanding technical requirements.

Kind regards,

Mohammad Nusair, Ph.D

Academic Editor

PLOS ONE
---

## [Editor Report · Acceptance letter]

PONE-D-25-05729R2

PLOS ONE

Dear Dr. Perraudin,

I'm pleased to inform you that your manuscript has been deemed suitable for publication in PLOS ONE. Congratulations! Your manuscript is now being handed over to our production team.

Kind regards,

on behalf of

Dr. Mohammad Nusair

Academic Editor

PLOS ONE